# Developing Complexity-Informed COVID-19 Responses to Optimize Community Well-Being: A Systems Thinking Approach

Stephanie Bogdewic [1] and Rohit Ramaswamy [1,2,*]

1   Gillings School of Global Public Health, University of North Carolina, Chapel Hill, NC 27599, USA;
    bogdewic@live.unc.edu
2   Cincinnati Children's Hospital Medical Center, Cincinnati, OH 45229, USA
*   Correspondence: rohit.ramaswamy@cchmc.org

**Abstract:** Despite a range of federal and state interventions to slow the spread of COVID-19, the US has seen millions of infections and hundreds of thousands of deaths. Top-down mandates have been ineffective because the community spread of the pandemic has been influenced by complex local dynamics that have evolved over time. Systems thinking approaches, specifically causal loop diagrams, and leverage points, are important techniques for representing complexity at the local level and identifying responsive systems change opportunities. This commentary presents a causal loop diagram highlighting the progressive effects of prolonged state-level COVID-19 mandates at the community level. We also identify potential system leverage points that address these effects and present an imagined future state causal loop diagram in which these solutions are implemented. Our future system demonstrates the importance of collaborations to enable community-driven, bottom-up approaches to public health crises, such as the COVID-19 pandemic, that are adaptive and responsive to local needs.

**Keywords:** COVID-19; systems thinking; causal loop diagram; leverage points



## 1. Introduction

As COVID-19 continues to evolve in the United States (US), it also rages on in other parts of the world, re-emerging in countries that had brought it under control. Within countries, the pandemic manifests itself in different ways in different locations, with contexts, behaviors, susceptibilities, and mutations intertwining in complex and unpredictable ways. With 178 million cases and 3.8 million deaths across the world by June 2021 [1], COVID-19 is a textbook definition of a "wicked problem" whose interacting dynamics make it impossible to define a single, universally applicable solution [2].

In 2017, the Centers for Disease Control and Prevention (CDC) introduced the *Community Mitigation Guidelines to Prevent Pandemic Influenza*, outlining key non-pharmaceutical strategies for slowing the spread of a virus [3]. For situations where a vaccine is not readily available, the CDC suggests a highly-coordinated, multi-pronged approach to rapidly employing individual safety measures (i.e., hand washing, mask wearing), preventing opportunities for community spread (i.e., social distancing, school and business closures), and encouraging routine cleaning of environmental surfaces [3]. Analysis of historical epidemiological data [4,5], as well as simulation models exploring pandemic influenza transmission [6,7], have consistently found that non-pharmaceutical interventions are most effective when employed rapidly at the beginning of a pandemic.

The US and most other countries failed to adopt community mitigation strategies in the early stages of COVID-19. As this initial opportunity was missed, the disease began to spread in complex and unpredictable ways. The spread of the disease exhibits what Peter Senge defines as "dynamic complexity" or situations in which:

*"cause and effect are subtle, and where the effects over time of interventions are not obvious . . . . When the same action has dramatically different effects in the short run and the long run, there is dynamic complexity. When an action has one set of consequences locally and a different set of consequences in another part of the system, there is dynamic complexity. When obvious interventions produce non-obvious consequences, there is dynamic complexity."* [8] (p. 71)

While our interconnected global systems contributed to the spread of the pandemic, national- and state-level policies intended to "flatten the curve" were enacted too late, and local dynamics took over. Despite federal and state action, 600,000 people have died in the US, and there have been upwards of 33.5 million cases nationally [1]. Disrupting health, education, and financial systems, COVID-19 has had a drastic impact on countless lives. There is a case to be made that after the initial opportunity for a coordinated response was missed, top-down mitigation responses are less useful because they are not sensitive to evolving local contexts. In a white paper published by the National Bureau of Economic Research, Desmet and Wacsziarg demonstrate that population density, age structures, and the percentage of the population in nursing homes are predictors of cases, disease severity, and deaths and advocate for local-level policy responses [9]. As we've seen over the past 18 months, state-imposed restrictions may have kept case counts down, but these restrictions resulted in a progressive deterioration in mental and social well-being of communities whose effects were not factored into state-level decision making. Despite the wide spatial heterogeneity across US counties, for a significant duration of the pandemic, most states treated every county in the state identically. Locally driven solutions to combat the spread of COVID-19, based on decision makers applying a systems thinking lens to understand the dynamics of COVID-19 in their own backyards, may have been a more holistic, well-being-focused approach [10,11].

Causal loop diagrams (CLDs) are useful visual tools for representing dynamic complexity in a system. In the early days of the pandemic, Sahin et al. used a CLD to depict the dynamics of the global pandemic and suggested leverage points to influence national and global policy [12]. In this commentary, we extend this analysis to show how CLDs can be used to represent how local dynamics at the level of a county in the US progress over time and affect physical, mental, and social well-being at the community level. We show the interactions of key economic, social, health, and environmental variables in a county as COVID-19 progresses and identify leverage points for change that can guide county-level actions and policies. CLDs were developed using Vensim PLE (version 8.2.0) software.

*Description of Setting*

This analysis is based on a "typical" US county whose demographic, economic, and infection data are close to both state and national figures. Comparative data are shown in Table 1 to provide insights into how this county represents a typical county and compares to the state and country data. The selected county has a 28.6% rural population [13], a median income of $49,688 [14], and the following breakdown of race: white (73.6%), Black or African American (20.9%), American Indian or Alaska Native (1.5%), Asian (1.7%), Native Hawaiian and other Pacific Islander (0.1%), and more than two races (2.3%) [14]. The county is primarily urban, and it has four metropolitan centers (populations ranging from 12,232 to 53,776) [15]. Fifty-six percent of the county's population is concentrated in roughly 12% of the land geography [15]. Of the state's 100 counties, 11 counties have two metropolitan areas (defined as populations greater than 10,000), four counties have three metropolitan areas, and five counties have four or more metropolitan areas—including the county used here [15]. There are six key employers within the county (employing more than 1000 people), including education, health services, private sector, and public administration; these employers account for roughly 23% of employment within the county [16]. During the 2020 Presidential election, 53.6% of the county voted for the Republican nominee, while 45.2% voted for the Democratic nominee [17]. Finally, among adult residents (over the age of 25) in the county, 86% have received a high school degree or higher, and 25% of adults

have received a Bachelor's degree or higher [14]. Compared to the country, this county is more rural, poorer, more diverse, and has a lower percentage of people with college degrees. Compared to the state, this county is less rural and more white. It, therefore, occupies a good "in-between" level relative to both the country and the state. The case fatality rate of COVID-19 in the county explored here (1.37%) is similar to the state-wide case fatality rate (1.32%) [1]. COVID-19 cases and deaths were counted from the beginning of the pandemic, or the first case in that location, to 15 December 2020—when the vaccine was rolled out in the US.

**Table 1.** Comparison of demographic, geographic, and economic data by county, state, and country.

|  | County | State | Country |
|---|---|---|---|
| Rural population | 28.6% [13] | 33.9% [18] | 19.3% [18] |
| Median income [14] | $49,688 | $54,602 | $62,843 |
| Race [14] |  |  |  |
| White | 73.6% | 70.6% | 76.3% |
| Black or African American | 20.9% | 22.2% | 13.4% |
| American Indian or Alaska Native | 1.5% | 1.6% | 1.3% |
| Asian | 1.7% | 3.2% | 5.9% |
| Native Hawaiian or Other Pacific Islander | 0.1% | 0.1% | 0.2% |
| Two or more races | 2.3% | 2.3% | 2.8% |
| Education [14] |  |  |  |
| High school graduate or higher | 86.3% | 87.8% | 88.0% |
| Bachelor's degree or higher | 24.8% | 31.3% | 32.1% |
| COVID-19 cases per 100,000 [1] | 5468 | 4257 | 5041 |
| COVID-19 case fatality rate [1] | 1.4% | 1.3% | 1.8% |

## 2. County-Level COVID-19 Dynamics

After the identification of a dozen confirmed cases of COVID-19 in the state, the Governor declared a state of emergency on 10 March 2020. By the end of the month, the Governor had issued various executive orders closing public schools, prohibiting gatherings, and promoting social distancing. The state-wide stay-at-home order was enacted on March 30, and individual counties implemented mask mandates throughout the initial stages of the pandemic. By May 2020, the Governor rolled out a phased reopening process, easing restrictions on retail stores and outdoor gatherings. A state-wide mask mandate was enacted by the end of June, as hospitalizations continued to rise. Within the county, there were tensions in how community members responded to the state mandates. Individuals expressed concerns with mandates at both ends—from protesting the reopening approach to expressing anger around masking and social distancing mandates. Businesses closures resulted in a rise in local unemployment. In response to these state-level mandates, county leaders took individual actions, such as developing a county recovery loan program for local businesses, writing a letter to the Governor requesting a regional approach to lifting restrictions, and local communication strategies to promote safe behaviors. These county-level initiatives, in response to both state-level mandates and the needs of the community, are typical of how local governments across the US have responded.

Figure 1 shows the CLD describing the interactions between the state and the county level as restrictions took effect and persisted. It is important to remember that the "loops" in a CLD are progressions over time and not static snapshots. To develop the CLD at the county level, we first explored key stakeholder perspectives by reviewing news and magazine articles published locally and nationally over the course of the virus in the US to determine the key stakeholders, their perspectives, and their interactions. At the same time, we reviewed the peer-reviewed literature for CLDs describing COVID-19 at the global or national level so that our county-level CLD could be linked to these higher-level dynamics where appropriate [12].

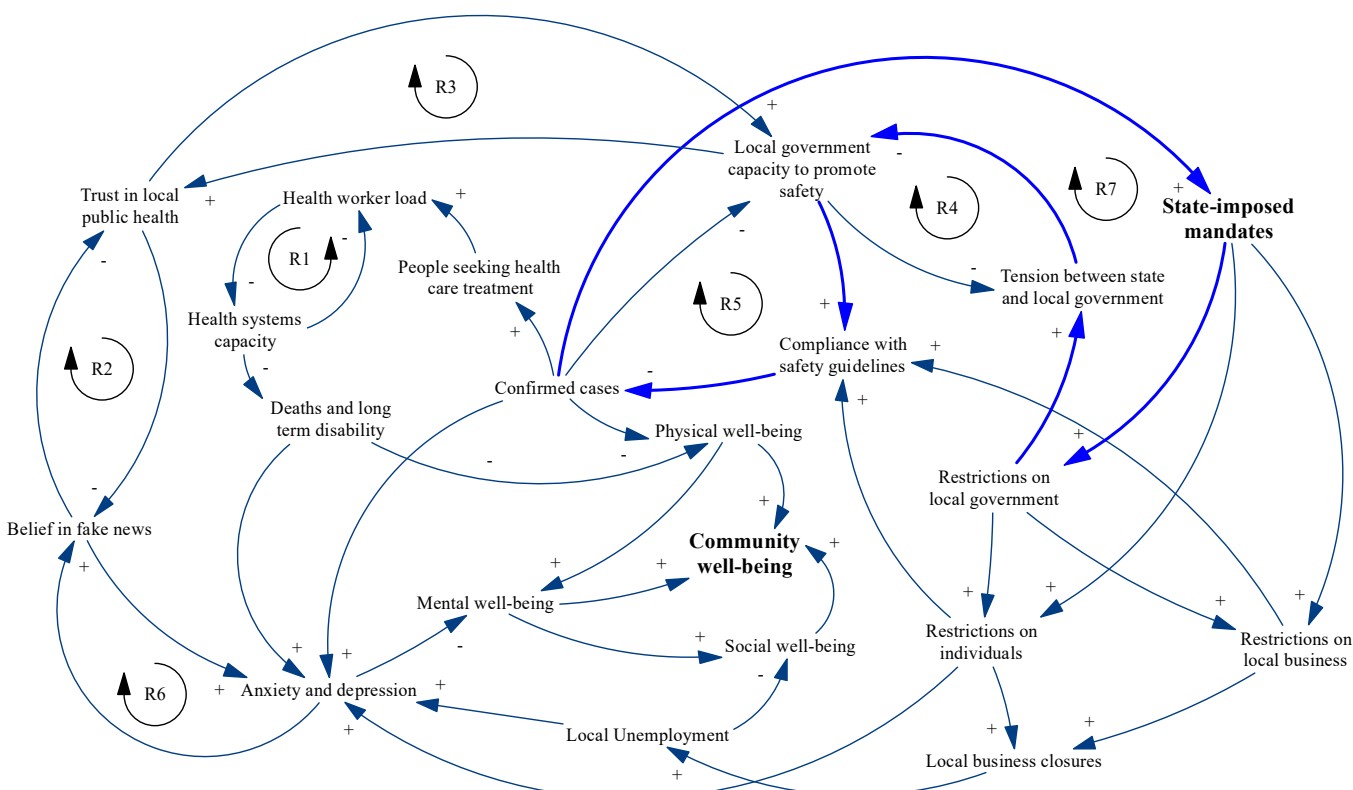

**Figure 1.** COVID-19 and community well-being causal loop diagram. Loop R7 is identified by lighter blue arrows.

The systems view represented by the CLD in Figure 1 explicitly recognizes the fact that COVID-19 has affected community well-being as a whole, which includes dimensions of physical, mental, and social well-being as recognized by the World Health Organization [19]. As shown in Figure 1 and described below in more detail, state mandates imposed universally on communities with the singular goal of reducing the impact of the pandemic on physical health affect mental health and social cohesion in varied ways. Figure 2a–c are subsets of Figure 1 illustrating these dynamics.

## 2.1. Impact of State Mandates on Community Physical Well-Being

For the purposes of this paper, we define physical well-being of the community as the absence of confirmed COVID-19 cases and related deaths. The effects of state-imposed mandates on this outcome are demonstrated in Figure 2a. Three mechanisms affect physical well-being. First, mandates lead to restrictions on local government, which manifest as restrictions on individuals (i.e., social distancing, mask wearing), which reduce spread and case counts. Second, as cases go down, the proportion of people seeking health care reduces, which in turn should reduce health worker load and increase health systems capacity (the number of beds and resources available to hospitals, as well as adequate availability of personal protective equipment, supplies, and other necessary equipment) as shown in loop R1. This increases the ability of health systems to treat serious cases and reduces disability and death. Finally, as cases go down, the capacity of local public health departments capacity to develop effective risk communication and improve outreach increases (loop R5). However, local health departments can also pay the price as a result of state mandates. If the mandates do not appear to be relevant to local authorities, state mandates can increase tensions between state and local governments. Local government authorities (such as sheriff's departments) could ignore or ineffectively implement state mandates, undermining compliance (loop R4). As a result, cases continue to rise, and the state doubles down with more mandates reinforcing the mistrust (loop R7). These

tensions have played out in many counties across the country during the course of the pandemic [20].

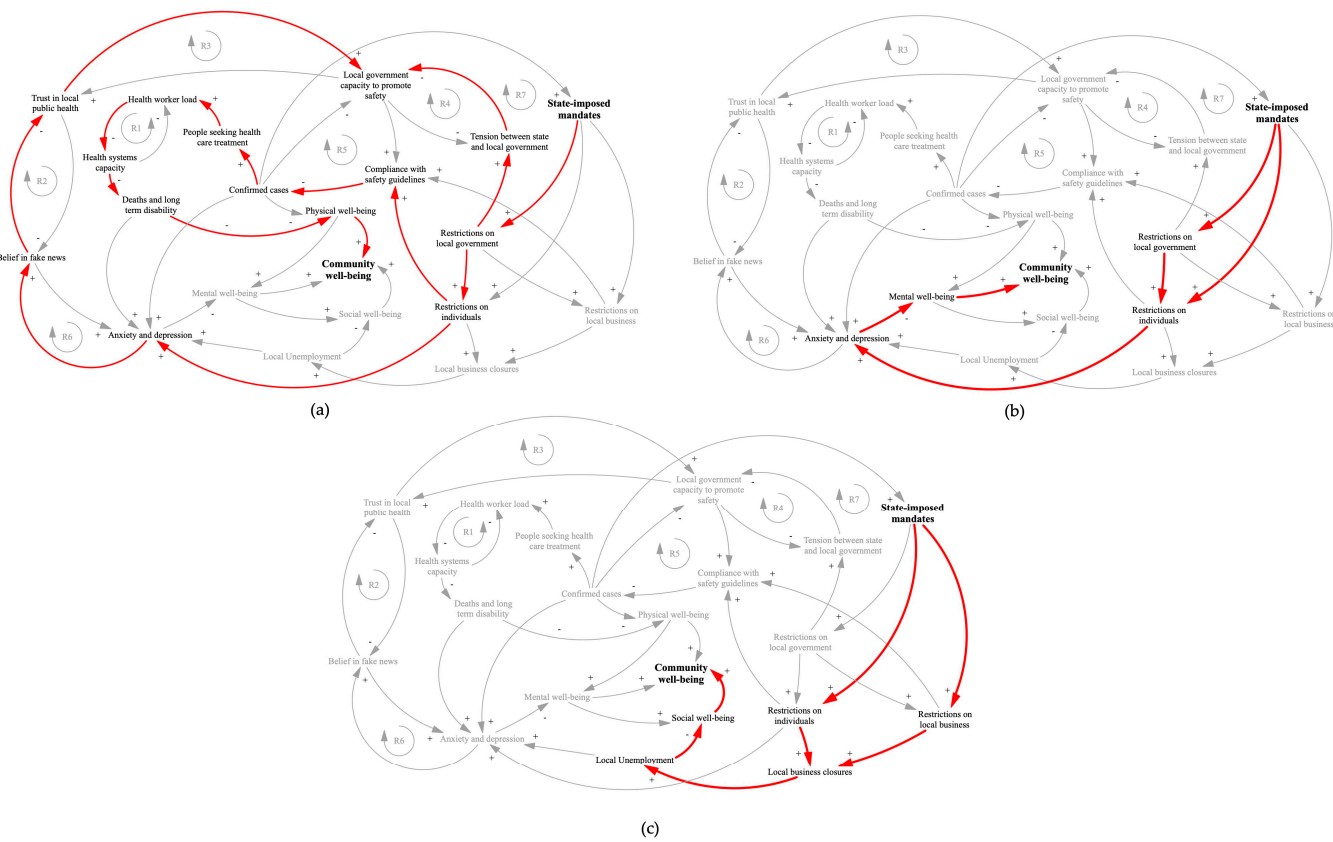

**Figure 2.** COVID-19 and community well-being causal loop diagram, subset dynamics (**a**) Pathway of the impact of state-imposed mandates on community physical well-being; (**b**) pathway of the impact of state-imposed mandates on community mental well-being; (**c**) pathway of the impact of state-imposed mandates on community social well-being.

### 2.2. Impact of State Mandates on Community Mental Well-Being

A property of dynamic complexity is that actions taken in one part of the system have consequences elsewhere. In the messy real world of US communities, the restrictions imposed by states do not immediately result in the anticipated reductions in cases or deaths. In response, many states double down and extend or increase restrictions, trying their best to "flatten the curve". As the restrictions persist, there are mental health consequences, demonstrated in Figure 2b. Community members become increasingly distanced and isolated and experience anxiety and depression resulting in a progressive decrease in community mental well-being. As a result, they may become more susceptible to the consumption of fake news that feeds on their fear and reinforces their anxiety (loop R6). Fake news also undermines trust in the public health system, which in turn affects public health's capacity to develop local responses to stem infections, which results in more restrictions and makes mental health worse (loops R2 and R3). Furthermore, deteriorated physical health may have a direct impact on decreased mental well-being, as indicated by the arrow from physical to mental well-being. In fact, the arrows in the CLD indicate the interrelationships among different forms of well-being, which is the emphasis of this paper. Physical well-being and mental well-being are related in complex ways, and in the system displayed in Figure 1, efforts to improve one aspect result in deterioration of the other.

Deterioration in mental health due to persistent COVID-19 restrictions is well known and has been widely reported. For example, prior to the COVID-19 global pandemic (January–June 2019), 11% of American adults ages 18 had reported symptoms of anxiety or depressive disorder [21]. Between the beginning of the pandemic and vaccine roll-out in mid-December 2020, reported symptoms of anxiety or depressive disorder peaked in early November at over 40% of the state's adult population [22].

### 2.3. Impact of State Mandates on Community Social Well-Being

We use the term social well-being to refer broadly to the well-being of a community, driven not only by the sense of connection between community members but also through their active participation in supporting the local economy by owning, working in, or patronizing businesses. As shown in Figure 2c, over time, as mandates restricting opening and closing of businesses persist or are reinforced and restrictions on individual mobility continue unabated, businesses close, unemployment at the community level increases, and the strength and vibrancy of the overall community decline. In addition to the direct effect, business closures also increase anxiety and depression, setting into play the vicious cycle of mental health deterioration described earlier.

## 3. Changing the Dynamics: Identifying Leverage Points

The CLD shown in Figure 1 exhibits characteristics of what is known in project management and evaluation as the "iron triangle" [23], in which it is not possible to simultaneously improve all three areas of well-being. This is because the system shown in Figure 1 does not represent a coordinated response that is aimed at reducing the spread of COVID-19 while providing support for mental health and minimizing economic distress. Rather, the system represents the top-down, one-size-fits-all approach that was adopted in the US and in many other countries, with a singular objective to reduce spread. While this might have been appropriate to reduce strain on the health system at the beginning of the pandemic, as we have demonstrated, over time, this created significant stress in other parts of the system and undermined the achievement of the primary objective.

This is not inevitable. We argue that local leaders, with knowledge of their particular contexts, can develop solutions in partnership with the state that address the local system's dynamics, leading to a deterioration of community well-being while accommodating state efforts to curb the spread of COVID-19. This can be done through the identification of *leverage points* which are places to intervene in a system where a small shift can lead to large-scale changes in system behavior [11]. The idea was first proposed by Donella Meadows, who identified twelve points ranging from small changes in systems parameters to shifting or transcending mindsets and paradigms [11]. The goal of leverage points is not to identify specific solutions but rather to help stakeholders consider where in a system it makes sense to intervene in order to disrupt dynamics that result in undesirable outcomes.

In Table 2, we present examples of leverage points that can be used to change the behaviors in Figure 1 that restrict community well-being. We have used Ehrlichman's categorization of ten leverage points into changes in four areas: *system infrastructure*, *information flows*, *organizing principles*, and *mindsets* [24].

**Table 2.** Description of leverage points and location within Figure 1.

| | Leverage Point | Location in Figure 1 | Descriptions |
|---|---|---|---|
| **Changes to System Infrastructure** | | | |
| 1 | Adding constraints | Compliance with safety guidelines | Enforcing restrictions on super spreader events |
| 2 | Changing rates | Local business closures | Reducing rates of local business closures by creating safe opportunities for businesses to interact with customers |
| 3 | Increasing buffers | Health worker load | Adding health worker resources to act as a buffer to increased patient volume |
| **Modifying Information Flows** | | | |
| 4 | Modifying feedback loops | R7 (Figure 1) to B1 (Figure 3) | Converting "state–county tension" to "state–county collaboration" |
| 5 | Expanding communication systems | Anxiety and depression | Increasing public communication channels through faith communities, community activists, local radio stations, and other trusted networks |
| **Redefining Organizing Principles** | | | |
| 6 | Changing the rules that govern the system | State-imposed mandates | Seeking to transfer authority for safety mandates to the county level stakeholders |
| 7 | Enhancing the organization of the system | Trust in local public health | Creating community partnerships to develop multifaceted response to COVID-19 |
| 8 | Aligning shared goals of the system | Community well-being | Acknowledging the need to simultaneously optimize physical, mental, and social well-being |
| **Altering Mindsets** | | | |
| 9 | Modifying the beliefs that guide behaviors in the system | Trust in local public health | Building community trust in local multi-stakeholder partnerships through communication and dialogue |
| 10 | Expanding the system's ability to transcend paradigms altogether | Community well-being | Creating opportunities for all stakeholders to engage in transparent dialogue about how to continually learn and adapt |

## 4. Using Leverage Points to Transform the System: A Thought Experiment

Application of the leverage points in Table 2 offers the possibility of shifting the system towards a balance of physical, mental, and social well-being. The modified CLD of this system is shown in Figure 3. The leverage points could bring about the changes resulting in the system of Figure 3 through the following two pathways described below.

### 4.1. Increasing Collaboration between State and Local Leaders

The impetus for this change is leverage point 8 that establishes a common goal among all stakeholders to focus on all aspects of well-being. To do this, instead of state-imposed mandates being automatic restrictions that must be enforced in a single inflexible way, the state sets up structures for ongoing state and local collaboration on the best way to communicate and enforce restrictions and to provide emergency support. For example, the state could collaborate with local authorities to create guidelines for outdoor spaces where people could be encouraged to visit safely and provide advice about community events that could bring people together in ways that supported local businesses. The state could help coordinate relationships between local government and hospitals to accelerate the provision of emergency supplies and resources when needed (leverage point 3). The primary idea is that the state continues to intensify collaboration with the county and builds its capacity to find locally acceptable solutions to promote safety (e.g., through leverage points 1 and 2), resulting in a reduction in case counts. The result is the opportunity described in leverage point 4. Decreasing tensions between the state and the county cause the conversion of the reinforcing loop R7 in Figure 1 (shown in light blue and representing an escalation of state actions in response to county-level case count increase) to the balancing loop B1 in Figure 3 (also shown in light blue and representing diminishing state action as local

capacity increases). The increased trust in local government encourages the creation of expanded messaging to combat the consumption of fake news (leverage point 5), resulting in reduced anxiety and improved mental health. Over time, as state and local authorities learn to work with each other, the state could divest decision-making authority about enforcing and easing restrictions to the local level (leverage points 6), further enhancing local capability to develop trust and support.

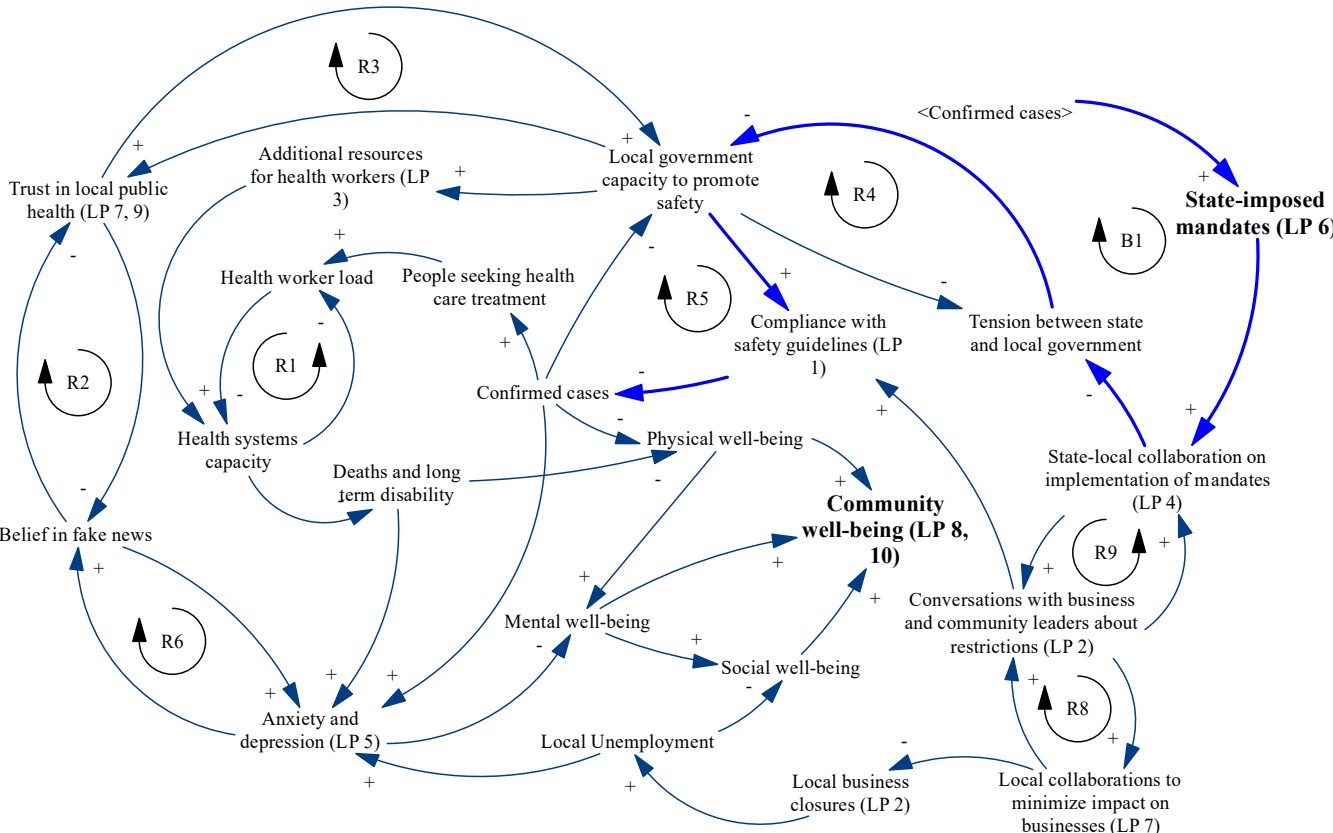

**Figure 3.** Future state community-level COVID-19 causal loop diagram, incorporating leverage points. Leverage point (LP) numbers from Table 2 are included in variable names; balancing loop B1 is identified by lighter blue arrows.

### 4.2. Increasing Local Multi-Stakeholder Community Partnerships

Leverage point 8 could also lead to the activation of leverage point 7, which could result in the creation of multi-stakeholder partnerships among business leaders, faith-based organizations, community coalitions, and local authorities in the form of COVID-19 response co-design teams. In the beginning, such teams could work on leverage points such as 2, 5, and 8, focusing on business support, communications, and trust building. As shown by the reinforcing loops R8 and R9 in Figure 3, these efforts, in conjunction with increased state–local collaboration, reinforce over time, positively affecting mental and social well-being. In the long run, there could even be the possibility of the complete elimination of state mandates, with the complete authority given to local partnerships to adapt, learn, and make decisions that are most appropriate for the local context. This is the "paradigm shift" described in leverage point 10 and represents the creation of local community capability to effect transformational change and address a variety of wicked problems, not just COVID-19.

## 5. Conclusions: A Call to Action

As the pandemic continues to evolve in much of the US, researchers are beginning to analyze our response to COVID-19 over the past year. In a recent paper, Vest and colleagues describe the challenges of translating national and state reopening plans into

local policy, stating that "*COVID-19 may be remembered as the disaster that created the zero-sum game between public health and the economy*" and reminding us that any plans for restrictions or reopening must pay attention to threats both to health and economic well-being [25] (p. 128). A National Bureau of Economic Research report by Agarwal et al. used data from 43 counties and all US states to conclude that Shelter in Place policies did not result in achieving the goals of reducing excess mortality [26]. One of the reasons they propose for their findings is a possible increase in "deaths of despair" resulting from social isolation and economic loss [26] (p. 6).

We must also recognize that the impacts of COVID-19 do not affect all individuals in the same way. While the dynamics represented in the CLD in Figure 1 might be experienced by everyone, the consequences of these dynamics vary widely. For example, within the county we have used as a setting for this CLD, employment decreased by 2.3% among individuals making more than USD 60,000 annually, compared to an 18.5% decrease among those making less than USD 27,000 annually [27]. There was a "risk divide" between those who are able to complete their jobs remotely and those who must be in-person, with a disproportionate number of Black and Hispanic individuals landing on the wrong side [28]. The household pulse survey conducted by the CDC showed significant variation in the prevalence of anxiety disorder by age, ethnicity, gender, and educational status [29]. Patterns of socialization in communities vary by culture, and the inability to gather with family and friends affects different ethnic groups in different ways. Additionally, empirical research has shown that there is variation in the characteristics of people who travel across counties—namely, socio-economic status, education, and population density—as it relates to the spatial interaction effect of COVID-19 transmission [30]. Depending on county-level characteristics, a prolonged state mandate to shut down all businesses or invariant social distancing rules will have different consequences in different counties. In the county we have used for illustration in this paper, employment decreased 15.5% between January and December 2020, compared to 10.3% in the state and 8.1% nationally [27]. Similarly, in the same period, the county saw a 32.4% decrease in the number of small businesses open, compared to 19.5% in the state and 28.9% nation-wide [27]. Any attempts to transform the system must explicitly include these equity considerations.

The overwhelming focus of the endless commentaries on COVID-19 responses in the academic literature and popular media has been on reducing case counts through top-down policy mandates. This commentary explores how systems thinking approaches and CLDs can be used to demonstrate the complex community-level interactions arising from these mandates and their effect on multiple facets of community well-being and postulates ideas for bottom-up community-led approaches to systems that are adaptive to community needs.

There is a well-established body of literature on the power of community coalitions to drive local-level transformation [31–34], but there is little knowledge about the use of these coalitions to address emergencies such as COVID-19. The common wisdom is that emergencies require a top-down response because resources and expertise exist at the federal or state level. While this may be appropriate when the emergency is localized (e.g., in the case of a natural disaster), distributed and adaptive approaches are needed when the emergency is widespread and heterogeneous. As we've seen throughout the country over the last year and a half, communities have found innovative ways to forge deep, personal connections and come together to take care of each other in a time of crisis [35]. However, these have been opportunistic and accidental and have taken place in spite of restrictions that have often been barriers to community efforts. Using the CLDs as a guide, we propose the creation of intentional multisectoral partnerships at the community level, facilitated by a stronger state-local collaboration that have the capability to respond adaptively and rapidly in times of crisis. These will require changes to established mindsets and paradigms. In her introduction to leverage points, Meadows ranked her twelve points in increasing order of effectiveness [11]. Changing mindsets and transcending paradigms are the most effective, but as Meadows indicates, these are the ones that the systems resist the most. Her

recommendation is for systems thinkers interested in bringing about change to "*model the system, which takes you outside the system and forces you to see it whole*" [11] (p. 18), and that is what we have attempted to do. We invite others involved at all levels of systems change to join in our common quest to seek solutions to COVID-19 and other wicked problems.

**Author Contributions:** Conceptualization, S.B. and R.R.; software, S.B.; writing—original draft preparation, S.B. and R.R.; writing—review and editing, S.B. and R.R. All authors have read and agreed to the published version of the manuscript.

**Funding:** This research received no external funding.

**Institutional Review Board Statement:** Not applicable.

**Informed Consent Statement:** Not applicable.

**Data Availability Statement:** No new data were created or analyzed in this study. Data sharing is not applicable to this article.

**Conflicts of Interest:** The authors declare no conflict of interest.

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
