# Peer review of "Developing Complexity-Informed COVID-19 Responses to Optimize Community Well-Being: A Systems Thinking Approach"

_systems, doi:10.3390/systems9030068_

Round 1

Reviewer 1 Report

This paper proposed optimization measures for community wellbeing based on the systems thinking approach of CLDs, which could provide a systematic perspective for counteracting COVID-19.  There are several points that might need revision or further explanation.

1 In section 2.1 the authors stated the potential tension between state and local government which leads to a worse situation, however, there is another situation in which the local government may disobey the reopening code for continuing strict regional mobility, such measures may improve the physical health of residents instead.

The quarantine code may affect the accessibility of daily health care, which may decrease the physical well being

2 In section 2.2 the mental wellbeing could be directly affected by physical well being, this may need to be added

3 In the section of Conclusion ,the authors pointed the limitation of their work, the situation varies according to SES, race, and education. the following paper may be helpful to further explain the spatial interaction effect of counties in states and relative factors.

The Spatiotemporal Interaction Effect of COVID-19 Transmission in the United States. ISPRS International Journal of Geo-Information, 2021. 10(6): p. 387.

Reviewer 2 Report

Line 69… 71.. “Causal loop diagrams (CLDs) are useful visual tools …………………….influence national and global policy [12]”.

Which software authors utilized to generate CLDs in the study? Whether authors have written their own script/developed tools or used any other software? Need to be clarified also provide proper references for third party application used (if any).

Reviewer 3 Report

The article is original, but has several important weaknesses. I include among them:

1. Derivation of general conclusions on the example of one country. The Authors do not name it, but state that it is "typical US country". It was worth selecting several different countries from different parts of the US for analysis, which would make the result more reliable.

2. The causal loop diagrams thoroughly prepared by the Authors do not allow, however, to prove the thesis that the bottom up method in combating the Covid 19 pandemic would be more effective than the top down method.

3. The Authors completely ignored two extremely important elements in the research. The first is the time factor - a quick response time to a pandemic threat is crucial. Consulting state officials with local communities on responses would extend this time, increasing the risk of spreading the virus. Secondly, pandemic restrictions are related to the use of coercive measures (especially in terms of the physical and social dimensions of well-being). State and federal authorities have the full control of such measures, which naturally makes them responsible for fighting the viral threat.

Despite the above-mentioned deficiencies, the main advantage of the article, which makes it recommended for publication, is that it is open-ended and initiates a discussion on the institutional aspects of combating the virus threat. Local authorities should be involved in this fight, but the question is how and in what model of relations with state and federal authorities. The article does not provide an answer to this, but initiates a discussion in this regard.
